# Telecommunication Facilities, Key Support for Data Management and Data Sharing by a Biological Mobile Laboratory Deployed to Counter Emerging Biological Threats and Improve Public Health Crisis Preparedness

**DOI:** 10.3390/ijerph18179014

**Published:** 2021-08-26

**Authors:** Aleksandr Vybornov, Omar Nyabi, Olga Vybornova, Jean-Luc Gala

**Affiliations:** Center for Applied Molecular Ttechnologies (CTMA), Institute for Clinical and Experimental Research (IREC), Université Catholique de Louvain, Tour Claude Bernard, Avenue Hippocrate, 54-55, bte B1.54.01, 1200 Bruxelles, Belgium; aleksandr.vybornov@uclouvain.be (A.V.); olga.vybornova@uclouvain.be (O.V.); jean-luc.gala@uclouvain.be (J.-L.G.)

**Keywords:** Biological Light Fieldable Laboratory for Emergencies (B-LiFE), biological threats, pandemic response, global health risks, remote locations, critical communications, TETRA, LTE, laboratory information management system (LIMS)

## Abstract

In the case of rapid outbreaks of infectious diseases in remote locations, the lack of real-time information from the field and rapid spread of misinformation can be a major issue. To improve situational awareness and decision-making at all levels of operational deployment, there is an urgent need for accurate, reliable, and timely results from patients from the affected area. This requires a robust and fast channel of communication connecting first responders on-site, crisis managers, decision-makers, and the institutions involved in the survey of the crisis at national, regional, and international levels. This has been the rationale sustaining the development of advanced communication tools in the Biological Light Fieldable Laboratory for Emergencies (B-LiFE). The benefit of terrestrial (TETRA, LTE, 5G, and Wi-Fi-Fi) and SatCom communications is illustrated through a series of missions and exercises conducted in the previous five years. These tools were used by B-LiFE operators to provide accurate, comprehensive, timely, and relevant information and services in real time. The focus of this article is to discuss the development and benefits of the integration of multi-mission, multi-user nomadic, rapidly deployable telecommunication nodes for emergency uses (TEN) in the capacity of B-LiFE. Providing reliable communication channels through TEN enables the development and use of an ICT toolbox called MIML_LIMS (multi-institution, multi-mission, multi-laboratory LIMS), a tool which is mandatory for efficient and secure data management and data sharing by a mobile laboratory.

## 1. Introduction

There are multiple international, European, and national (Member States’) stakeholders and coordination mechanisms that are involved in a health crisis, such as an epidemic or pandemic outbreak (http://www.pandem.eu.com, accessed on 25 August 2021) (Figure 1) [1,2,3,4]. The given overview in Figure 1 uses a top-down approach, starting from the international level and then moving to the European and then national levels [4].

An important task of first responders and/or public protection and disaster relief (PPDR) organizations and services, such as healthcare workers, medical teams, police, firefighters, civil protection, and defence experts, is to deal with emergency and surveillance situations in different activity areas (e.g., health care, civil protection, defence) in any part of the world [1,4,5,6,7,8,9]. The most significant activities, which are proposed in this work, are carried out on-site in the field, so all the tools match the needs and requirements of the practitioners and customers.

In the field of first response to disasters or public protection and disaster relief, radio communications are extremely important systems that enable people to communicate. Telecommunication systems are currently connected to data and media, allowing all stakeholders to share information in a timely manner, including relief organizations that are rapidly deployed on-site. We know now that crisis or disaster situations can happen in any part of the world or even become pandemics, as evidenced by the COVID-19 disease. In most acute crisis situations, radio communication remains the only available communication tool.

As illustrated in Figure 2, the generally agreed categories of emergency communication are [10]:Communication between authorities/organizations. This category refers to communication within and among authorities/organizations, and fits within the scope of PPDR communications;Communication from authorities/organizations to citizens. This category refers to communications from authorities/organizations with individuals, groups, or the public. Warnings and information systems to alert the population are part of this category;Communication of citizens with authorities/organizations. Emergency call services (e.g., calls to emergency numbers, such as 112 through public telephone networks) are part of this category;Communication among citizens. In the case of a disaster, individuals may have a strong demand to communicate among themselves e. g; to state or determine the status of relatives, property as well as to coordinate actions of mutual interest. New social media communication technologies can potentially enable citizens to share information faster, assist in response and recovery during emergencies, and mobilize for action in political crises.

Based on the PPDR-specific communication needs as presented above, Section 2 describes the progressive integration of a communication capacity in the B-LiFE deployable (box-based) laboratory. In this section, we illustrate and discuss the integration process implemented through a few main field deployments and exercises focusing on rapid response to biological crisis. We explain how efficiency, robustness, and sustainability of the integrated telecommunication and box-based laboratory concept was assessed and how it was used to ultimately improve the performance of the B-LiFE team during operational deployments.

## 2. Humanitarian Deployments and International Exercises

### 2.1. Deployment in Guinea

Following the dramatic evolution of the Ebola crisis in West Africa, Belgium and Luxembourg have contributed to the national and international stakeholders (WHO, West African countries authorities, etc.) who require assistance. A joint deployment of a light-fieldable analytical capacity and a logistic chain of support for providing rapid diagnosis of Ebola viral disease (EVD) was considered the best support, and it could be used by the local Ebola treatment units. At the request of the government of Guinea, a B-LiFE (Biological Light Fieldable Laboratory for Emergencies) “Ebola mission” was organized with the support of the Belgian Civil Protection and Belgian Defence.

The B-LiFE/B-FAST (Belgian First Aid and Support Team) mission was deployed in N’Zerekore, Forest Guinea, from 20 December 2014 to 22 March 2015 [11]. This mission actively contributed to the international fight against the spread of EVD in West Africa. The Belgian B-LiFE/B-FAST intervention was performed in three rotating shifts of teams consisting of three biologists and a medical doctor in charge of the mission, from the Centre for Applied Molecular Technology (CTMA/IREC/UCLouvain) and a logistical support cell with four people (a decontamination expert from the civil protection service, an officer in charge of security and safety aspects, a military expert in satellite communication, and a military nurse).

The mission was supported by B-FAST during the first two months and then jointly by the Belgian Technical Cooperation and the European Commission (DG ECHO).

B-LiFE and the Emergency.lu service provided by SES TechCom and the Luxembourg Government enabled the laboratory to have an outstanding satellite communication capability, allowing for secure communications at high speeds with the Belgian and international operational centres. The generated results were transmitted by satellite and stored in a secure central database in Belgium.

The satellite telecommunication capacities of B-LiFE were indispensable in supporting the deployed biological analytical capacity of the work. The head of the B-LiFE mission could have real-time video conferences with the laboratory in N’Zerekore and medical experts in Belgium, providing real-time expert advice to help interpret the analytical results and quickly adapt patients’ treatment when required.

The Emergency.lu C-band satellite terminal provided by SES TechCom (Betzdorf, Luxembourg) and the government of Luxembourg was deployed for the B-LiFE mission for three months and performed efficiently during this time (Figure 3). The 2 MHz bandwidth SES satellite capacity was sufficient for the communication needs of the mission. Real-time satellite communications were found to be critical for:Mission management and communication with the home base;Logistics management (acquiring consumables, reagents, spare parts for lab equipment, etc.);Communication with stakeholders (WHO, ECDC, Belgian, Guinean authorities, etc.);Transferring the results to secure servers;Video conferences with remote experts;Video conferences with the families of the lab team for psychological support.

No other fieldable laboratory deployed during the Ebola epidemics had these satellite telecommunication capacities.

### 2.2. Deployment in Germany

On 8–12 February 2016, B-LiFE was deployed in Munich [12], side by side with the military deployable capacity of the Bundeswehr Institute of Microbiology (IMB), which operates the EU Mobile Labs and has also been deployed for the Ebola crisis in West Africa.

This exercise was called Clueless Snowman and aimed to compare the B-LIFE civilian and IMB military mobile capacities for procedures and performance, as well as their preparedness for responses to waterborne diseases. Water-related emergency preparedness and outbreak responses have become significant issues in recent history, such as natural disasters (floods, hurricanes, or droughts), man-made disasters (intentional contamination), and outbreaks (infections linked to the exposure of soiled water).

B-LiFE was deployed in the fictive context of a natural disaster affecting the sanitary conditions of a rural region in Germany. The fictitious poor sanitary conditions had led to outbreaks of diseases due to water contaminated with pathogens. The Belgian B-LiFE lab and German IMB Bio-lab performed sampling and analysis of the samples using next-generation sequencing (NGS) technology in the field.

For the genomic analysis and on-site identification of unknown agents, a next-generation sequencing device called MinION sequencer, produced by Oxford Nanopore Technologies (ONT) Company (Oxford, UK) (Figure 4), was used to characterize and identify emerging pathogens. Satellite communication assets allowed the transfer of raw data from the sequencing device to a remote server. This asset was also crucial for discussing operational issues locally or with distant stakeholders. Furthermore, telecommunications were essential for the quick diffusion of the analytical results.

The following critical advantages of the satellite communication were also tested and validated during the exercise:Patients’ clinical data were fully protected and transmitted to secure servers;Real-time video conferences to provide help from remote experts in the treatment of patients was available when required;Real-time contact with the home base and stakeholders/beneficiaries (including WHO, ECDC, UN, Belgian ministries, etc.);Without satellite communications, remote experts could not be reached in a timely manner; therefore, the local exchange with the ability to generate communication sessions for two or more parties is something that was not possible in laboratories without SatCom (Figure 4);Map data could be refreshed and updated maps can be integrated into B-LiFE in real time to provide decision support to the head of the B-LiFE deployment team, B-LiFE home base team, and stakeholders/beneficiaries;Fast logistics support responses (additional equipment, space parts, reagents, etc.);Without SatCom, conventional missions could require 20 or more additional days to receive the required reagents or equipment;Remote permanent audio and video contacts between the volunteers deployed in the field and their families by various messenger applications such as Skype, WhatsApp, FB messenger, etc., will be key to preserving the moral spirit of the mission, which was not possible before or was very limited due to the poor-quality and expensive short cellular phone conversations.

The 2 MHz bandwidth of the SES satellite capacity (based on the standard service license agreement (SLA)) was sufficient for the communication needs of the mission, even at peak hours. In the future, and to enable the transfer of an increased amount of analytical data to the storage server, it will be necessary to increase the satellite bandwidth capacity.

### 2.3. Deployment in Sweden

The European Commission endowed the Falck company (https://www.falck.com/en/, accessed on 6 March 2021) with different tasks relating to the design, planning, conduction, and evaluation of five exercises for the Medium/Heavy urban search and rescue (USAR), USAR in chemical, biological, radiological, nuclear (CBRN) conditions, Advanced Medical Post with or without surgery, Field Hospital, Technical Assistance Support Team (TAST), European Union Civil Protection Team (EUCPT), and other capacities of the Voluntary Pool.

The fourth exercise of this cycle featured medical modules and Bio Mobile Labs and was held in Revinge, Sweden, on 24–27 April 2017 [13]. The exercise MODEX 2017 was organized, on behalf of the European Commission, for Emergency Medical Teams (EMT), Mobile Laboratories, EUCPT—experts of the European Civil Protection Mechanism, and TAST. The exercise focused only on medical teams. As such, it provided the only opportunity for medical teams in the cycles of exercises to cooperate with each other and to test their interoperability and procedures.

B-LIFE facilities were deployed under the flag and support of the Belgian First Aid and Support (B-FAST).

Following the scenario of this exercise, the southern area of the Faultland region suffered severe flooding. Alerts were raised concerning diseases that may have spread throughout the region. Retrospective studies were conducted by WHO staff, and health officials identified the index case in Lund. The exact source of the infection was not identified. A pattern of unprotected exposure, more cases and deaths, more funerals, and further spread has been established.

The Ministry of Health of Faultland issued its first alert for an unidentified disease. Following investigation by the Faultland Health Ministry, Lund and Eslöv have been identified as the epicentres of disease transmission that was caused by an unknown agent. The WHO was alerted to the emergence of a “rapidly evolving” outbreak in this region. Mobile laboratories were then deployed in the area to support local health agencies with the testing and confirmation of cases.

In addition, Faultland was suffering from the collapse of the health system: access to basic and emergency care was challenged by infrastructure damage and lack of medical personnel. They were unable to take care of the injured as a direct and indirect result of flooding. Faultland, therefore, addressed a request for assistance to the EU Civil Protection Mechanism.

The following modules were deployed on the affected Faultland:The Italian Advanced medical post with surgery (Type 2);The Spanish Field Hospital (Type 2);B-LiFE under B-FAST command and support;The German EU Mobile Lab;The Austrian Technical Assistance and Support Team;SES TechCom provided two satellite telecommunication assets (Figure 5);GATR inflatable antenna plus NoSaCo terminal operated in C-band via SES-5 satellite (Loral Space & Communication, New York, NY, USA) and based on the emergency.lu platform;Astra2Connect fast deployable kit operated in Ku-Band via the Astra-3B satellite and based on the Astra2Connect platform.

Both satellite assets were combined using a load balanced broadband router, and the results showed that the total throughput of the satellite backbone was 19 Mbit/s (DL)/6 Mbit/s (UP), and the latency was approximately 700 ms, which is a good result for this type of equipment. This solution was very robust, despite one of the satellite links being out of order, as another continued to operate. The increased throughput of the backbone ensured the transfer of large volumes of data during the exercise (Table 1) and enabled an outdoor WiFi access point to allow end-users to access the internet in the field.

### 2.4. Deployment in Belgium

The Bio-Garden exercise was held on 19 June 2018 at the Belgian Defence barracks Major Housiau at 181, Martelarenstraat 1800, Vilvoorde (Peutie), Belgium, and was organized jointly by the H2020 project eNOTICE—European Network of CBRN Training Centres (Grant Agreement No. 740521, https://www.h2020-enotice.eu/, accessed on 18 August 2021), UCL/CTMA team, and the ESA B-LIFE telecommunication consortium [14,15].

The Bio-Garden exercise was the result of a close collaboration between the Ministries of the Interior, Defence, and Public Health, and several Belgian and European SMEs, aiming to improve the preparation and response of first responders in case of a bioterrorist attack.

The European Commission (DG HOME, DG ECHO, and DG DEVCO) and the European Space Agency fully supported this joint international exercise by providing high visibility to first responders, decision- and policy-makers, and by attending the exercise.

The general context of the Bio-Garden exercise was Level 3 in European security applicable to EC countries, justified by an unprecedented wave of terrorist activity in several European cities in the preceding months. In June 2018, the weather was unusually hot and “Nowhere Country” experienced an outbreak of norovirus-based gastrointestinal infections in the population.

Following an informer’s tips, the police forces feared a biological attack and actively searched for evidence of terrorist activity. As one of the preventive measures for the final of the European Champion Leagues 2018 in the capital city, the “Nowhere Country” authorities decided to preventively deploy two analytical capacities (a B-LiFE deployable laboratory and a Hungarian military deployable laboratory) near the football stadium. Both capabilities were in stand-by mode and were ready to operate. Based on the exercise scenarios, the accident scene of the exercise is shown in Figure 6.

Bio-Garden took place on the morning of 19 June 2018 and had two tightly interconnected and simultaneous parts:Investigations were conducted in a clandestine laboratory; the police discovered a clandestine laboratory containing recipients with white powder and coloured liquids, which allowed us to suspect the manipulation and/or production of bioweapons. Several response teams (police, civil protection, and military CBRN sampling teams) intervened in the investigations, providing samples to the deployed laboratories, and reporting to federal authorities (Federal Crisis Center and its CBRN Coordination Center).Food defence activities: At the same time, severe gastrointestinal symptoms were reported in VIP persons invited the day before to a lunch in one of the capital’s best restaurants, as part of the social events preceding the football final. The public health authorities were informed that the same symptoms affected several citizens who had ordered pizzas the previous evening. All pizzas have apparently been delivered by the same pizzeria “No Name Bio-pizza”.The outline of the scenario underlined:Interaction and information sharing between police, civil protection, and military CBRN sampling teams in the clandestine lab;The role of both deployed laboratories for the fast analysis of samples received from both the clandestine lab and the suspicious food samples (from the restaurant and the pizzeria);The importance of communications, that is, timely, relevant and well-coordinated information sent to the Federal Crisis Centre and its new CBRN Coordination Centre, and the speed of the feedback information to the scene.

To support the participants of this exercise, the B-LiFE telecommunication team prepared a new solution (fast-deployment telecommunication node for a biological light-fieldable laboratory for emergencies) based on the new mobile communication platform provided by SES (including satellite and WiFi subsystems), terrestrial telecommunication subsystems, based on the TetraNode platform provided by Rohill (Hoogeveen, Netherland) and Nokia Ultra Compact LTE solution (Espoo, Finland).

The generic architecture of a fast-deployment telecommunication node for a biological light-fieldable laboratory for emergencies is shown in Figure 7.

The B-LiFE telecommunication team installed an omnidirectional antenna for TETRA and a directional antenna for LTE on fast-deployment antenna mast#1 (H = 3 m) near the B-LiFE telecommunication tent. Three-directional WiFi antennas (to provide better coverage) with the outdoor WiFi APs and the antenna for point-to-point (PtP) configuration were installed on antenna mast#2 (H = 3.5 m), and the counterpart was installed on fast-deployment telecom mast#3 (H = 3.5 m) near the Civil Protection TAST and C2 Bus location. The SES satellite link was used as the backbone for the TetraNode platform. The SES TechCom provided a rapidly deployable satellite capacity in the Ku- and C-Band, which was accessible on-site through the SES rapid response vehicle and the “zero tools” Gigasat deployable antenna (Figure 8).

During the exercise, the Ku-Band capacity was configured for a symmetrical 2 Mbps/2 Mbps service rate, and the 2 MHz C-band capacity was connected according to the standard SES SLA. With this combination of both satellite assets, a total throughput of approximately 5 Mbit/s in Uplink/Downlink (UL/DL) was achieved.

Users/stakeholders in the field had access to the internet resources (laboratory information management system, epidemiological mapping application, etc.) via SES TechCom Managed Wireless (WiFi) Communications that covered the area of operations.

The main characteristics of the solution were to ease the deployment, ensure scalability, and provide flexible user management.

The unique communication resources management features ensure that the WiFi network provides a positive user experience while maintaining applications at the necessary level of service.

In the scenario, exercise communication between two mobile labs (Belgian B-LiFE and Hungarian lab) was supported by two-way radio communication through the Rohill fast deployable TetraNode system.

The reason for the communication between the two labs was that the pathogen found by the Hungarian lab in the pizza/oysters had to be proved to be exactly the same as the pathogen discovered by the Belgian B-LiFE lab in the samples from the clandestine lab. Deep sequencing (MinION) is the only way to determine this, and this is highly relevant for FCC and for justice; if the pathogens were exactly the same, then the food intoxication with the pizza and oysters would have been proved to be ipso facto from a criminal/intentional origin.

Furthermore, communication between the B-LiFE lab and the clandestine lab was also provided by the Rohill fast deployable TetraNode system (Figure 9). In this case, the Sampling and Identification of Biological, Chemical and Radiological Agents (SIBCRA) team assessed the cleanliness of the front door (face inside the lab) using a new BC sense sensor developed by Estonian SME LDI Innovations. The task was to monitor the residual contamination on the door (a special contaminated area was provided and stuck to the surface of the door). A swab sample was collected when a positive signal was produced. This sample was then placed on a drone that transported the sample to the B-LiFE lab. The sample for transportation by the drone was only a test, as current legal procedures forbid the transportation of samples in such a manner, but technologically it is possible. The drone pilot was located close to CPC2.

TETRA/LTE PTT services were also demonstrated during the *Bio-Garden* exercise on 19 June 2018. The SES satellite link was used as a backbone for these services, and the backbone was used to interconnect smartphones with the Rohill TeamLink application to the TETRA radios in the field.

TeamLink is the Rohill suite of applications for real-time voice, status, text, location, photo, and video delivery. TeamLink is available as an app for Android and iOS smartphones as well as client software on Windows PCs, and it operates as an over the top (OTT) application on top of 3G and LTE networks. OTT refers to the transparent delivery of voice and data over a fixed IP, WiFi, 3G, or 4G wireless connection without the need for services in the core infrastructure in order to control and charge for this functionality.

The TeamLink solution was designed to support Mission critical push-to-talk (MCPTT) in the upcoming releases of the TeamLink Core and TeamLink for Android/iOS software. This ensures that TeamLink users receive a guaranteed future-proof solution that works with both private and public (operated) LTE networks based on open IP and LTE standards.

On the server side, TeamLink was supported by the TeamLink Core solution. The critical voice and data protocol (CVDP) are the basis for the operation of TeamLink. The TeamLink app can be provisioned for the redundant operation of the TeamLink Core, allowing virtually seamless switchovers when link or equipment failures occur. Its operation is also much faster and more robust in comparison with current push-to-talk (PTT)-over-cellular and proprietary solutions with a central call manager.

The TeamLink app provides an intuitive user interface; voice, status, text, and future photo and video capabilities are accessible by using a few buttons and gestures that are available on the main screen. Most of the functionality is invisible to the user and is indicated only when appropriate.

To implement the *Bio-Garden* exercise scenario, it was important to provide real-time voice and video data for the SIBCRA team. This was done using the Nokia push-to-video application with an LTE smartphone camera (Nokia Ultra Compact Network LTE system; Figure 10) and a SIBCRA staff member to transmit voice, image, and video two-way.

The Belgian B-LiFE mobile lab had direct discussions with the SIBCRA team during the sampling procedure. The SIBCRA team also transmitted real-time video to the B-LiFE lab. The SIBCRA person inside the clandestine lab exchanged useful information the B-LiFE team on the visual observations made to better define the types of possible criminal activities carried out directly inside this facility.

Prior to the exercise, preliminary calculations for the radio coverage of the TETRA and LTE systems were performed. The recommended ITU-R P.525-3 was used as the propagation model for the calculation.

For all terrestrial networks TETRA and LTE coverage planning was performed for the uplink direction (device to base station) as it is the limiting factor. For the TETRA network, the design was made for a typical voice call service, and for the LTE, the design was made for a 1 Mbps data service under the worst conditions (cell edge).

Corresponding to these scenarios, different link budget calculations were performed. The key parameters are listed in Table 2.

The average calculation results for the communication range are presented in Table 3.

During the exercise tests, drives using the Keysight measurement equipment (Nemo Outdoor solution) were also compared with the preliminary calculation results.

### 2.5. Deployment in Italy

The COVID-19 pandemic is creating a catastrophic health situation worldwide and in all European countries. However, some European countries are suffering more than others. In spring 2020, Italy had the largest number of victims after China. The Italian health infrastructure was consequently overloaded. This increased the need for reliable diagnostic tests for suspected cases, as a critical measure to reduce the spread of the disease.

With the support from the European Space Agency, B-LiFE was deployed from 11 June to 24 July 2020, in Italy, in the Piedmont Region, to bring additional diagnostic test capabilities, including real-time quantitative polymerase chain reaction (RT-qPCR) analysis and rapid detection tests based on lateral flow assay to determine the prevalence of the COVID-19 disease in the first responders using serological test, QuickZen COVID-19 IgM/IgG (ZenTech^®^, Baltimore, MD, USA) [16] (Figure 11). Moreover, it was also utilized to train local biologists [17].

B-LiFE deployed its multi-mission, multi-user telecommunication emergency node (TEN; Figure 12) integrating a range of new technologies, including satellite telecommunication and terrestrial communication (LTE, TETRA, and WiFi enabling broadband and narrowband communications simultaneously), mapping capabilities for situational awareness and epidemiological maps, and an autonomous information management system, including a laboratory information management system (LIMS) and a screening tool for patients.

The mission was supported by the Luxembourg government that provided the high-performance capabilities of the GovSat-1 satellite, Institute Pasteur—Paris (France), with biologists who were integrated in the B-LiFE-deployed teams and the company ZenTech (Liège, Belgium), which supplied COVID-19 serologic tests.

To increase the resilience and total throughput of the satellite connections, satellite terminals were combined using a load balance broadband router. LTE, TETRA, and WiFi subsystems were simultaneously provided broadband and narrowband communications. The LTE and TETRA subsystems provided interconnections with existing telecommunication networks or with the internet in the case of deployment in a remote area.

Furthermore, an SES Network Scorpion MP80X terminal and two Skynet terminals were also used.

#### 2.5.1. Terrestrial Telecommunications

The following main activities were carried out during the mission:using the French SME ETELM tactical system to provide voice communication (individual, group, and emergency calls);using the LTE part of the ETELM tactical system to provide local data transfer;using the satellite link to interconnect the local system with the internet.

As the LTE system (Figure 13) was mainly used around the lab tent and the C2 tent, the antenna mast for the LTE transmitter was installed near the C2 tent. For the TETRA system, however, the priority was outdoor coverage, and consequently, another antenna mast was installed near the satellite dish.

Before switching on the ETELM equipment, spectral analysis was performed with an Anritsu S412E Radio Analyzer to ensure that the frequencies had not been used and to prevent any potential risk of interference.

Private radio networks (TETRA and mission critical LTE) are valuable assets in critical situations such as pandemics or biological/chemical emergencies. The ETELM’s mission critical radio solution helped to ensure efficient and reliable communications in the field, enhance the situational awareness of the B-LiFE mobile lab users, and achieve greater operational performance during emergencies.

#### 2.5.2. SatCom Communications

The B-LiFE mobile laboratory relied on an end-to-end, satellite-enabled connectivity solution put in place by the SES and built on previous successful projects under the ESA Space Solutions framework and the company’s extensive expertise in the emergency and humanitarian domain. The Luxembourg Department of Defence, together with GovSat (Luxembourg), supported the mission in Piedmont by leveraging portable terminals as well as the high-performance capabilities of the GovSat-1 satellite operating in the X-band (the outdoor arrangement of the Scorpion MP80X terminal is shown in Figure 14).

The solution based on a lightweight portable manpack enabled to secure 10 Mbps duplex connectivity for health data processing, real-time data transmission, and communication with remote experts.

The Eutelsat Ka-band innovations enabled the world’s highest capacity Ka-band satellites to deliver the best broadband internet speeds around the world (the arrangement of 2 Tooway SatCom dishes is shown in Figure 15). Residential terminals include an attractive indoor unit (IDU) and an unobtrusive outdoor unit (ODU) that enables fast web browsing, video streaming, file sharing, and bandwidth-intensive internet applications.

Uplink up to 10 Mbps;Monthly allowance 500 GB;Reduced speed rate (when the full allowance is consumed);Downlink up to 3 Mbps;Uplink up to 1 Mbps;One public fixed IP provided;Internet access provided.

#### 2.5.3. Speed Tests

During the mission, different configurations of the SatCom and 4G/LTE equipment were tested and validated to achieve the maximum IP backbone bandwidth. The speed test results of the 4G/LTE modem, SES Scorpion MP80X terminal, and Eutelsat SatCom kits 1 and 2 are shown in Figure 16, Figure 17, Figure 18 and Figure 19.

The best results were achieved after combining all the aforementioned facilities using a load balance broadband router. The speed test results after this combination are presented in Figure 19.

#### 2.5.4. Laboratory Information Management System (LIMS)

The use of a tactical telecommunication node (TEN) bubble provides the possibility of integrating all data in a tracking and tracing ICT system called a laboratory information management system (LIMS). The LIMS solution selected according to the requirements of the B-LiFE was the LabCollector software suite developed by the French Company AgileBio, Paris, France with a patient registration database developed by the Belgian SME Eonix (Mons, Belgium). The LIMS was used and adapted during the mission through continuous interactions between AgileBio and Eonix and B-LiFE operators working on-site.

While patient registration was carried out by Italian first responders through the Eonix IT interface, the LIMS LabCollector enabled the B-LiFE team to encode patient analyses and the corresponding results, as detailed below. It should be noted that the LabCollector was originally designed for large stationary labs, but its modularity and adaptable architecture allowed the partners to adjust the original design of the tools and modules to the specific constraints and requirements of B-LiFE. Programming work was carried out in partnership with the AgileBio team to make the LIMS system compatible with the B-LiFE laboratory activities and the objectives of the mission.

The main functionalities needed during this mission were patient data collection, including the physical addresses of the patients for epidemiological mapping, generation of analysis requests (serology (IgG and IgM antibodies) and molecular (RT-qPCR) tests), and the generation of sample analysis reports. Two types of reports were generated: a portable document format (PDF) printable report and a global result file was produced daily, which integrated the totality of the results generated in the B-LiFE laboratory for transfer into the Italian national health reporting system.

The LIMS specificities developed during the B-LiFE Piedmont mission allowed for optimal management of patient data, but also efficient and user-friendly management of the results of the tests performed. This system enabled the B-LiFE team to quickly encode a large amount of data, in particular through the “batch” function available in the LabCollector. Moreover, in only a few days, data communication was fully compatible with the Italian national notification system.

Using LIMS in the field presumes the collection and storage of patients’ personal data. Personal data protection and data security issues are taken most seriously and are addressed with the highest priority by a LIMS developer, and by the laboratory staff collecting personal individual data, in full compliance with the European General Data Protection Regulation (GDPR; https://gdpr.eu/, accessed on 12 July 2021). The LIMS software ensured secure data storage at the level of every system file, as well as regulated access to the data only for authorized users. The patients’ personal data were shared only with the accredited health authorities of the host country. If requested by any other stakeholder in any other location, for example, for statistical purposes, all the personal data were filtered out, providing only pseudonymized data, making it impossible for external stakeholders to identify a patient but enabling the authorized staff members to correct possible encoding errors. At the same time, the data transmission channels themselves were secured to prevent personal data from leaking.

#### 2.5.5. Epidemiological Maps

Satellite images for maps can be obtained from multiple databases, including COPERNICUS. The COPERNICUS Emergency Management Service can be activated if necessary.

There are four types of maps embedded in the LIMS:Global maps, which give an overview of all relevant spatial data for the mission;Epidemiological maps, which provide a summary of the analysed patient data visualizing clustered patient information per administrative/geographical region;Samples/alerts/patient maps, which visualize the sample/alert/patient information in real time as it is collected on the terrain;Mobile-friendly version of the sample/alert/patient map.

The epidemiological map generated during the mission in Piedmont contains the following basic functionalities: search by location; print map; zoom to overview; check scale map; check coordinates mouse position.

The epidemiological map summarizes all patient data in the districts selected as areas of interest (Figure 20).

The epidemiological map contained extra functionalities to alter the content on the map and select a specific time window. The summarized patient information was visualized for each district. District borders were provided at the community level. New district borders were added automatically to the map for patients originating from new areas. The summarized patient information was then visualized. The number of patients examined, as well as “cured cases”, was visualized on the map.

## 3. Conclusions

Modern telecommunication assets are critical for any mobile biological laboratory, especially when they are deployed in response to a public health crisis occurring in a remote area or in areas where the existing telecommunication infrastructure is partly or totally lacking or not functional.

These telecommunication facilities provide mobile laboratory staff with the possibility of supporting operational duties using voice and video calls, as well as the ability to exchange e-mails and messages, transfer laboratory data, and access remote servers and databases from any part of the world.

The lessons learned from successive missions, as well as national and international exercises, demonstrates the increasing role of wideband telecommunications, primarily based on LTE, in addition to existing conventional and/or narrowband radio communications, such as TETRA systems. The increasing use of on-site next-generation sequencing methods to follow up the genetic features of a targeted pathogen, as well as the diagnostic technologies and tools used to enhance situational awareness, such as drone images, has led to a rapid increase in the size of data that must be transferred to relevant stakeholders, remote servers, and databases through the cloud.

These fast-growing communication constraints prompt telecommunication engineers, whose role is to support the mobile laboratory capacity with sufficient communication resources, to consider the implementation of wideband telecommunications based on 5G standard as soon as possible. In this respect, high-throughput satellite services are also very promising [18].

Considering the positive results obtained during previous deployments such as the Ebola mission in West Africa in 2014–2015, and the success of the recent Italian anti-COVID-19 mission, and taking advantage of the extensive experience in combining laboratory and telecommunication tools, the leading researchers of the CTMA laboratory launched the independent company Global Mobile Laboratory (GML, Saint-Hubert, Belgium; https://globalmobilelab.com/, accessed on 29 July 2021). The GML objective is to improve the laboratory operational capacity by continuously integrating innovations with the support of projects funded by the European Commission and the European Space Agency. GML, which consists of CTMA staff members and mission partners, has the necessary knowledge and competence to implement ambitious research and business applications.

The major focus of new GML activities in the telecommunications field is to continue the development of multi-mission, multi-user nomadic, rapidly deployable telecommunication nodes for emergency uses (or telecommunication emergency node (TEN)). This solution will be presented as an “all-in-one” (AIO) form-factor/conception, defined as a fully integrated stand-alone solution that provides all types of required telecommunication services, including terrestrial (TETRA, LTE, 5G, and WiFi) and SatCom communications for PPDR end-users/stakeholders, irrespective of the type and location of the crisis. For example, tactical telecommunication bubbles will provide coverage to the users at ~10 km with TETRA, ~1 km with LTE, and ~0.2 km with WiFi and 5G, depending on the landscape (flat, hilly, or wooded) and deployment scale (small/medium/large). The number of potential users supported by TEN was estimated to be approximately 25/50/100 at the three scales, respectively.

This telecommunication infrastructure will provide reliable channels for developing and using an ICT toolbox called MIML_LIMS (multi-institution, multi-mission, multi-laboratory LIMS). It will be fully configurable manually or via web services and fully integrated with cartography in real time for data analysis. This ICT toolbox also contains mapping and real-time situation awareness modules. All ICT applications will use the secure infrastructure of the European Distributed Data Centre developed and supported by Eonix.

## Figures and Tables

**Figure 1 ijerph-18-09014-f001:**
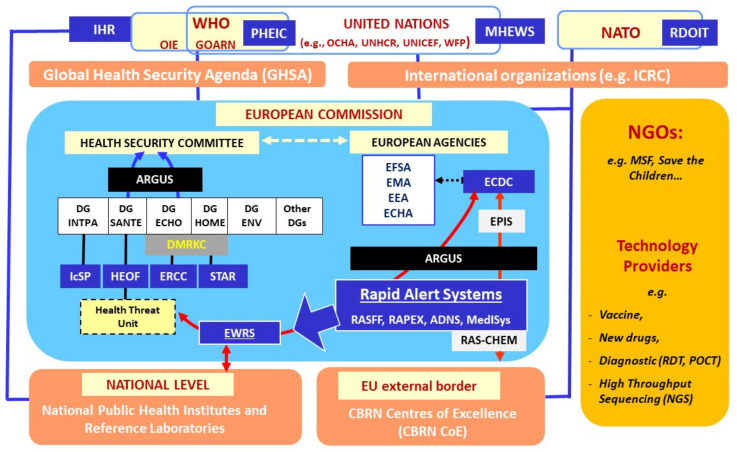
Overview of the health crisis stakeholders’ interactions and alert mechanisms at the international, European, and national levels. ADNS: Animal Disease Notification System; ARGUS: Alerte Rapide, Globale et Sûre, the internal general European rapid alert system, which is an internal crisis management mechanism of the Commission; DG: Directorate-General–Internal Partnerships (INTPA); Health and Food Safety (SANTE), Environment (ENV), Humanitarian Aid and Civil Protection (ECHO), Migration and Home Affairs (HOME); CBRN CoE: Chemical Biological Radiological and Nuclear Risk Mitigation Centres of Excellence; DMRKC: Disaster Risk Management Knowledge Centre, which supports the translation of complex scientific data and analyses into usable information and provides science-based advice for DRM policies; ECDC: European Centre for Disease Prevention and Control; EEA: European Environment Agency; EFSA: European Food Safety Authority; EMA: European Medicines Agency; EPIS: Epidemic Intelligence Information System, an alert and risk assessment system for threats of a biological origin; ERCC: Emergency Response Coordination Centre; EWRS: Early Warning and Response System, a rapid alert system for notifying alerts at the EU level on serious cross-border threats to health; GOARN: Global Outbreak Alert and Response Network; HEOF: Health Emergency Operation Facility, which coordinates the management of public health emergency at the EU level; ICRC: International Committee of the Red Cross; IsSP: Instrument contributing to Stability and Peace, the largest European civilian external security programme; IHR: International Health Regulations, an international legal instrument that covers measures for preventing the transnational spread of infectious diseases; OCHA: Office for the Coordination of Humanitarian Affairs; OIE: World Organization of Animal Health; MediSys: Medical information system; MSF: Médecins Sans Frontières (Doctors without Borders); PHEIC: Public Health Emergency of International Concern, a formal declaration by WHO of “an extraordinary event which is determined to constitute a public health risk to other States through the international spread of disease and to potentially require a coordinated international response”; POCT: point-of-care tests; RAPEX: Rapid Exchange of Information System, and alert system for unsafe consumer products and consumer protection (non-food products); RASCHEM: Rapid Alert System for chemicals; RASFF: Rapid Alert System for Food and Feed; RDT: Rapid Diagnostic Tests; RDOIT: Rapidly Deployable Outbreak Investigation Team, which aims to investigates outbreak(s) or incidents(s) where the intentional use of biological agents (biowarfare, bioterrorism, or biocrime) cannot be excluded; NATO: North Atlantic Treaty Organization; UNHCR: United Nations High Commissioner for Refugees; UNICEF: United Nations Children’s Fund; WHO: World Health Organization; WFP: World Food Programme.

**Figure 2 ijerph-18-09014-f002:**
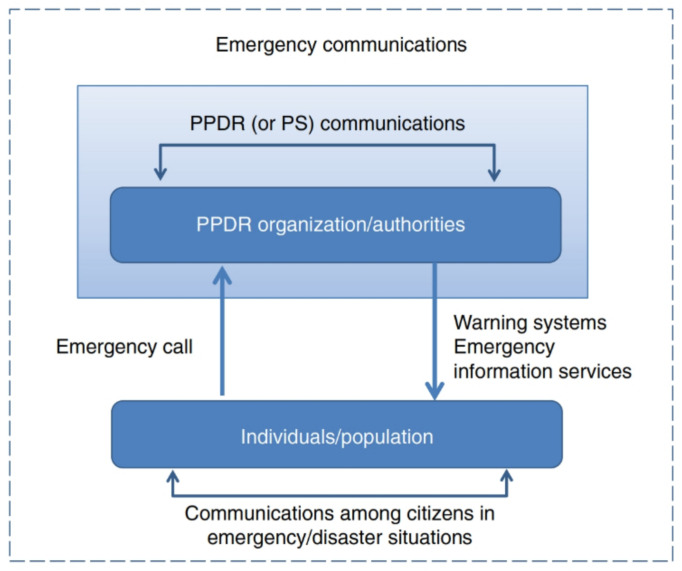
Scope of Public Protection and Disaster Relief (PPDR) and emergency communications.

**Figure 3 ijerph-18-09014-f003:**
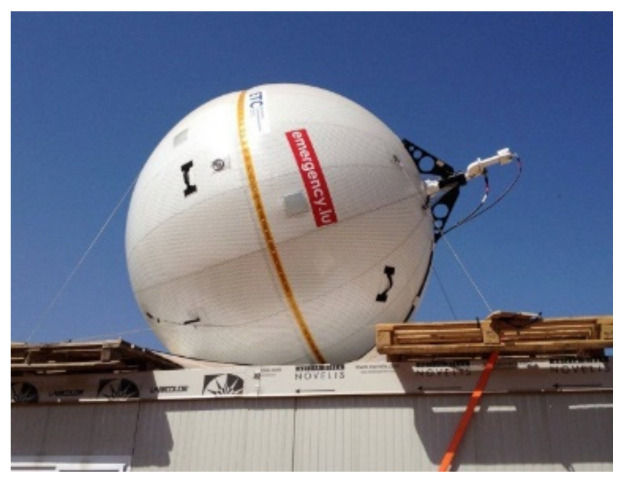
B-LiFE deployment in N’Zerekore, Guinea: view of the satellite communication antenna (2.4 m inflatable Ground Antenna Transmit and Receive (GATR); www.gatr.com, accessed on 17 February 2021, obtained from–SES-Société Européenne des Satellites, Betzdorf, Luxembourg) placed on the roof of the B-LiFE command and control shelter.

**Figure 4 ijerph-18-09014-f004:**
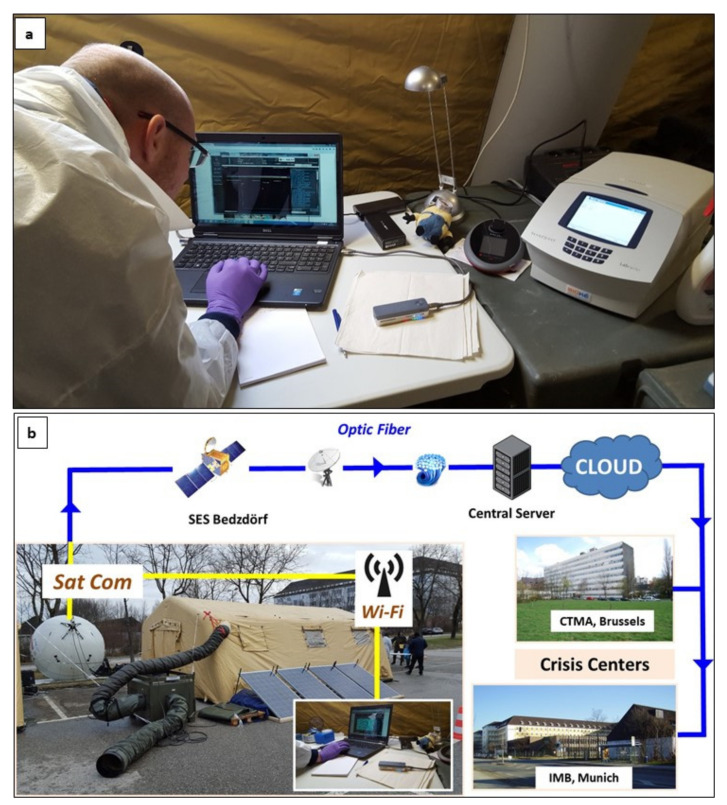
Exercise “Clueless Snowman” in the Institute of Microbiology, Bundeswehr, Munich: (**a**) Use of the MinION sequencer (ONT) in the B-LiFE facility during the exercise “Clueless Snowman”; (**b**) MinION data flow from the laboratory to the tent to the crisis centre.

**Figure 5 ijerph-18-09014-f005:**
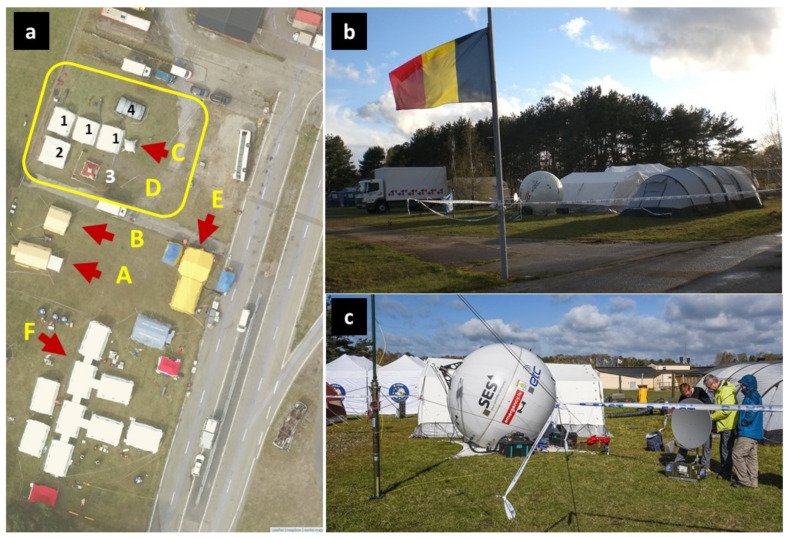
B-LiFE/B-FAST deployment during the MODEX international exercise in Sweden, 20–24 April 2017. (**a**) Aerial view by drone: arrows indicated the laboratory tent with the sample reception area (**A**), the SatCom command and control post (**B**), the GATR antenna for SatCom (**C**). The yellow square (**D**) shows the B-FAST (Belgian first aid and support team) logistic complex with the tent set-up comprising a dormitory (**1**), a dining area (**2**), an infirmary (**3**), and a global command and control tent (**4**). Arrows **E** and **F** indicate the Italian Field Hospital (**E**) and the EUCPT tent (**F**). (**b**) Global view of the B-LiFE/B-FAST deployment site. (**c**) Inflatable GATR and dish antenna for satellite communication.

**Figure 6 ijerph-18-09014-f006:**
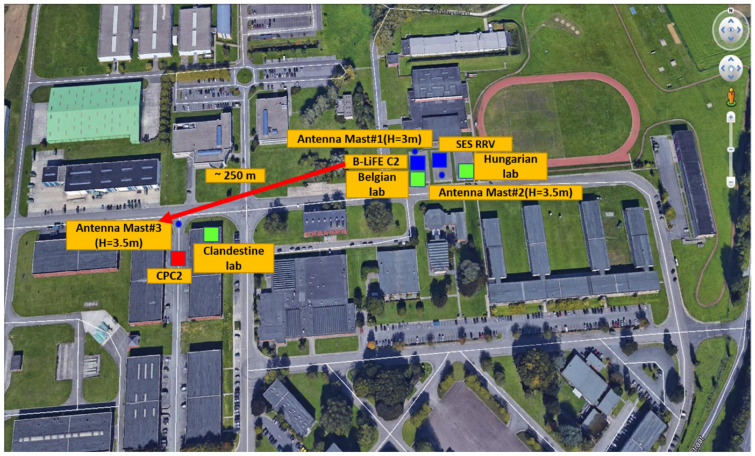
Global view of the “*Bio-Garden*” Exercise, conducted in the military domain Major Housiau, Vilvoorde, Belgium. This international exercise t” ok place in the framework of the implementation of both ESA/ IAP/ARTES B-LiFE and Horizon 2020 eNOTICE research projects.

**Figure 7 ijerph-18-09014-f007:**
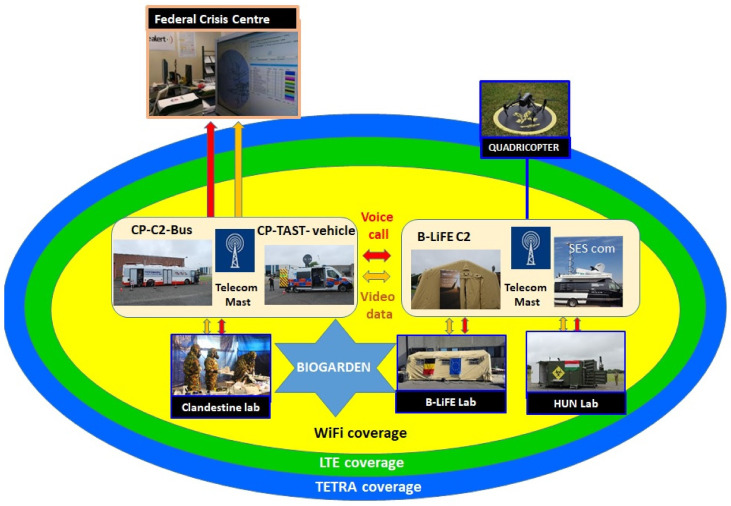
*Bio-Garden* exercise: Schematic representation of the *Bio-Garden* participants and telecommunication tools.

**Figure 8 ijerph-18-09014-f008:**
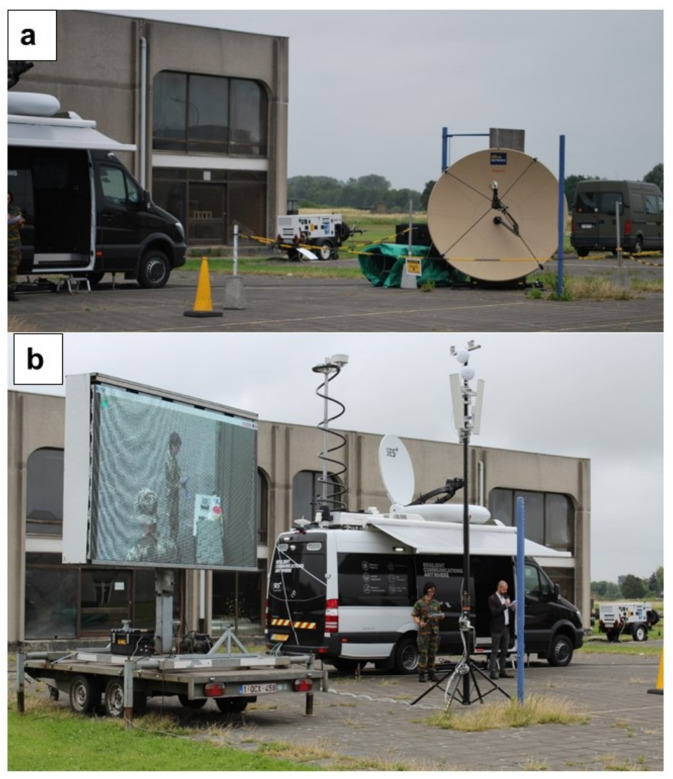
*Bio-Garden* Exercise: (**a**) Gigasat deployable antenna and antenna mast#2 with WiFi equipment; (**b**) SES rapid response vehicle.

**Figure 9 ijerph-18-09014-f009:**
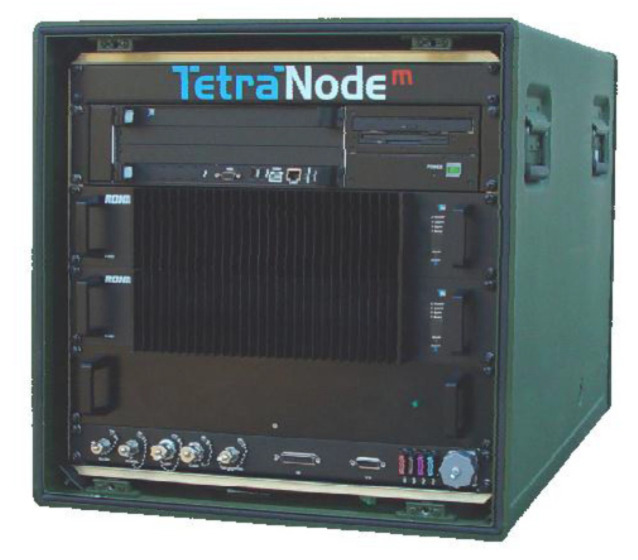
*Bio-Garden* exercise: Rohill TetraNode equipment (TETRA base station) tested during the exercise.

**Figure 10 ijerph-18-09014-f010:**
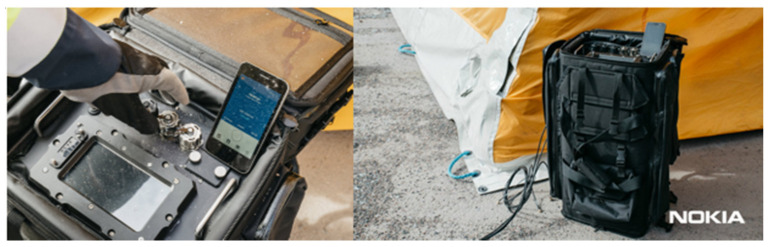
*Bio-Garden* exercise: Nokia Ultra Compact Network LTE system (LTE portable base station) tested during the exercise.

**Figure 11 ijerph-18-09014-f011:**
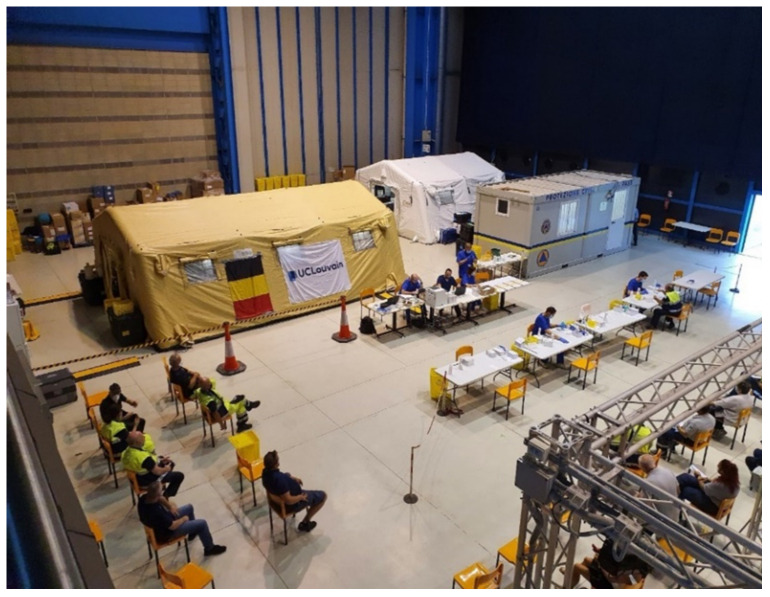
Overview of the B-LiFE deployed in Turin, Italy, from June to July, during the COVID-19 crisis.

**Figure 12 ijerph-18-09014-f012:**
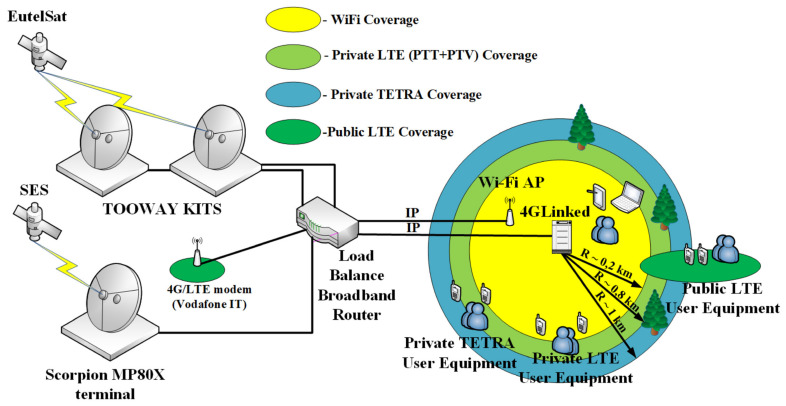
B-LiFE deployment in Turin, Italy, during the COVID-19 crisis: set-up of the telecommunication emergency node (TEN) diagram (equipment architecture and functioning).

**Figure 13 ijerph-18-09014-f013:**
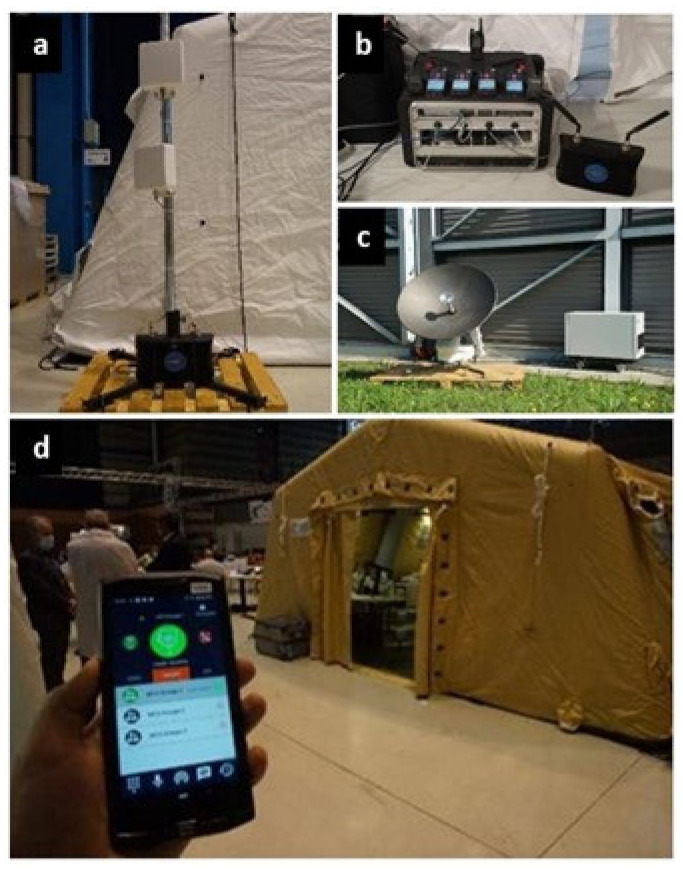
B-LiFE deployment in Turin, Italy, during the COVID-19 crisis: ETELM equipment (TETRA and LTE base stations and subscriber terminal) used for the terrestrial telecommunication. (**a**) Installation of the LTE eNodeB; (**b**) the nodal (core) system with user terminals; (**c**) installation of the hybrid TETRA BS near the dish antenna of the satellite link; (**d**) LTE terminal with an ongoing group call diagram.

**Figure 14 ijerph-18-09014-f014:**
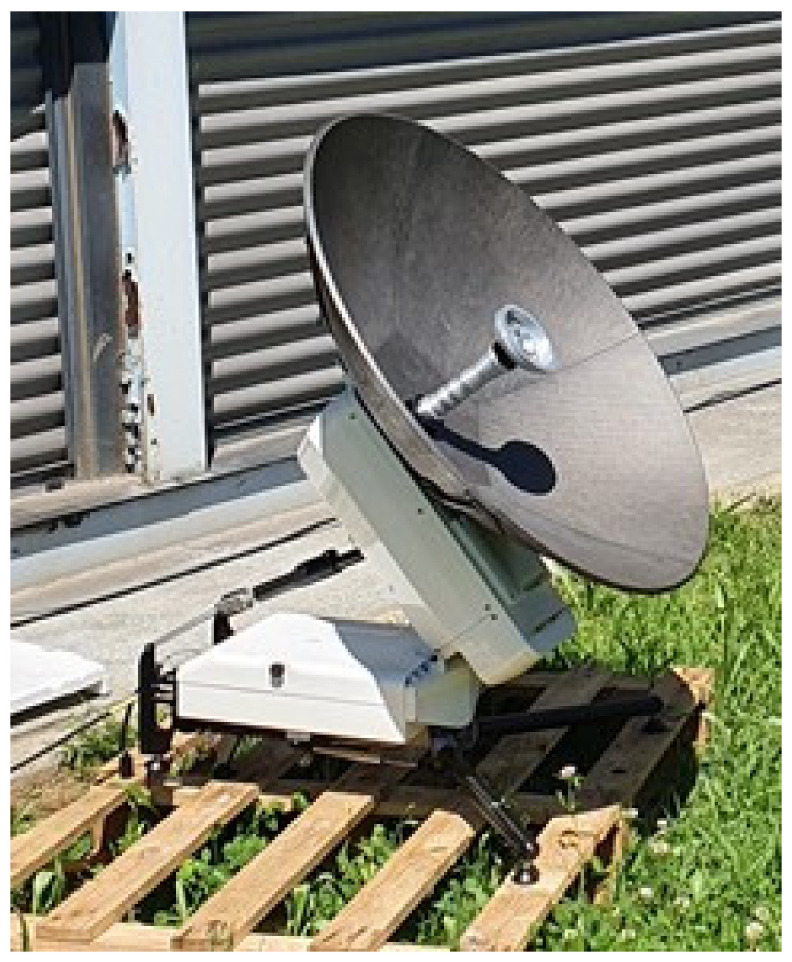
B-LiFE deployment in Turin, Italy, during the COVID-19 crisis: installation of the GovSat Scorpion MP80X SatCom terminal.

**Figure 15 ijerph-18-09014-f015:**
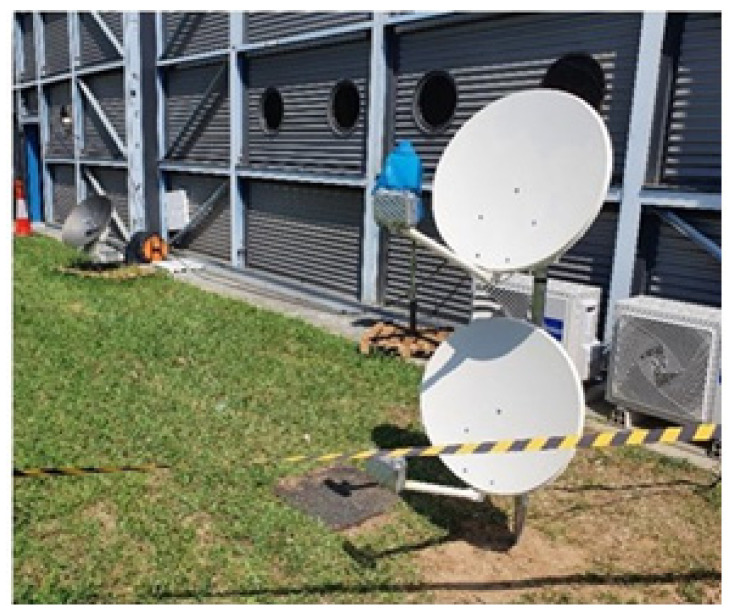
B-LiFE deployment in Turin, Italy, during the COVID-19 crisis: installation of 2 Eutelsat SatCom dishes.

**Figure 16 ijerph-18-09014-f016:**
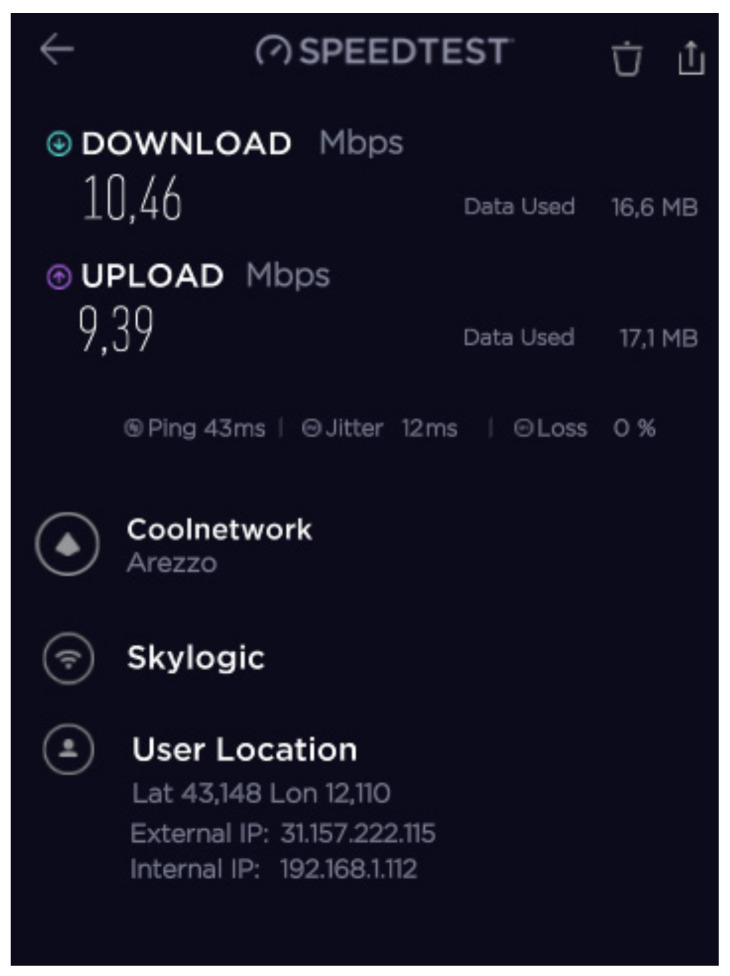
B-LiFE deployment in Turin, Italy, during the COVID-19 crisis: speed test results for the 4G/LTE (Vodafone IT) modem.

**Figure 17 ijerph-18-09014-f017:**
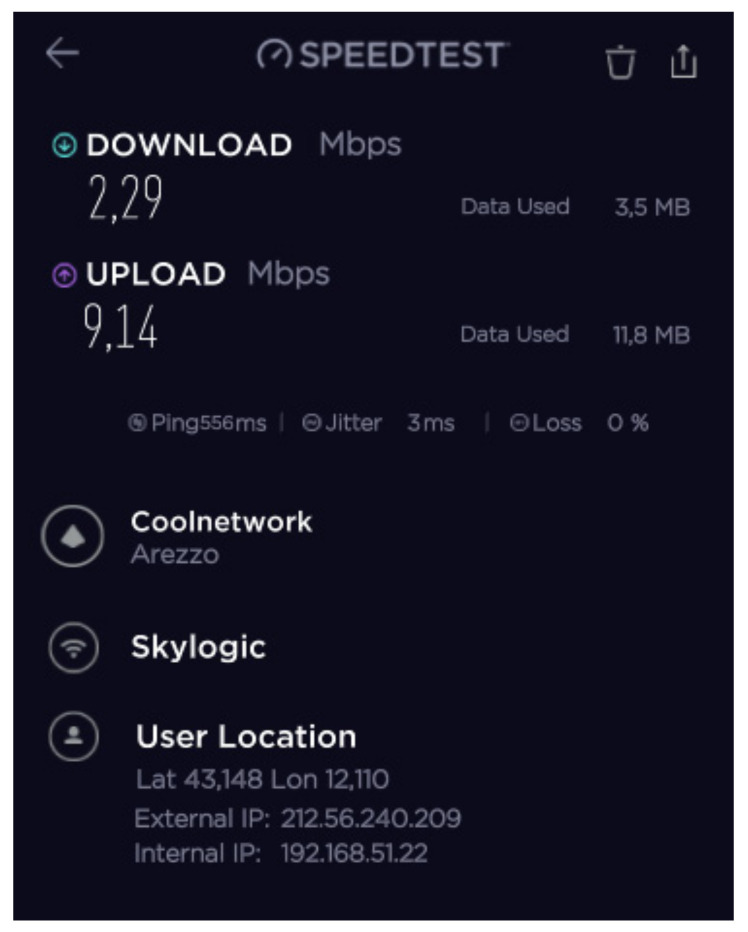
B-LiFE deployment in Turin, Italy, during the COVID-19 crisis: speed test results for the SES Scorpion MP80X terminal.

**Figure 18 ijerph-18-09014-f018:**
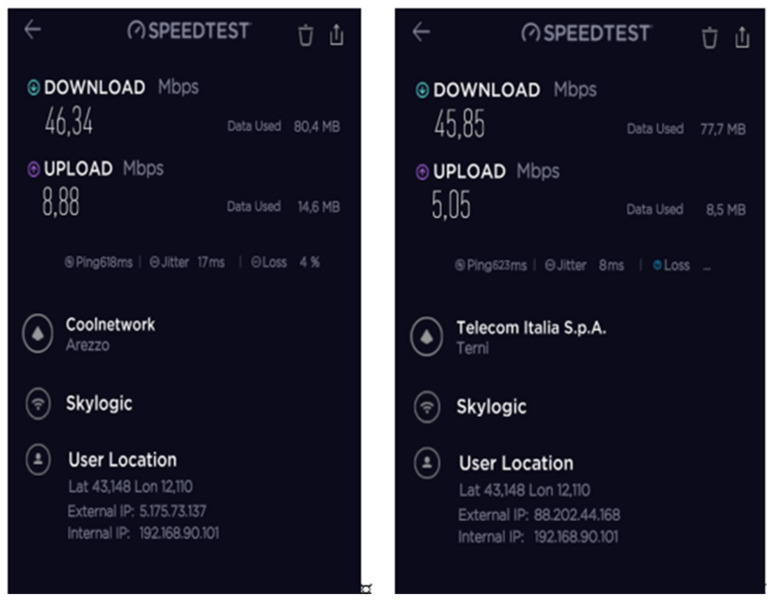
B-LiFE deployment in Turin, Italy, during the COVID-19 crisis: speed test results for the Eutelsat SatCom kits 1 and 2.

**Figure 19 ijerph-18-09014-f019:**
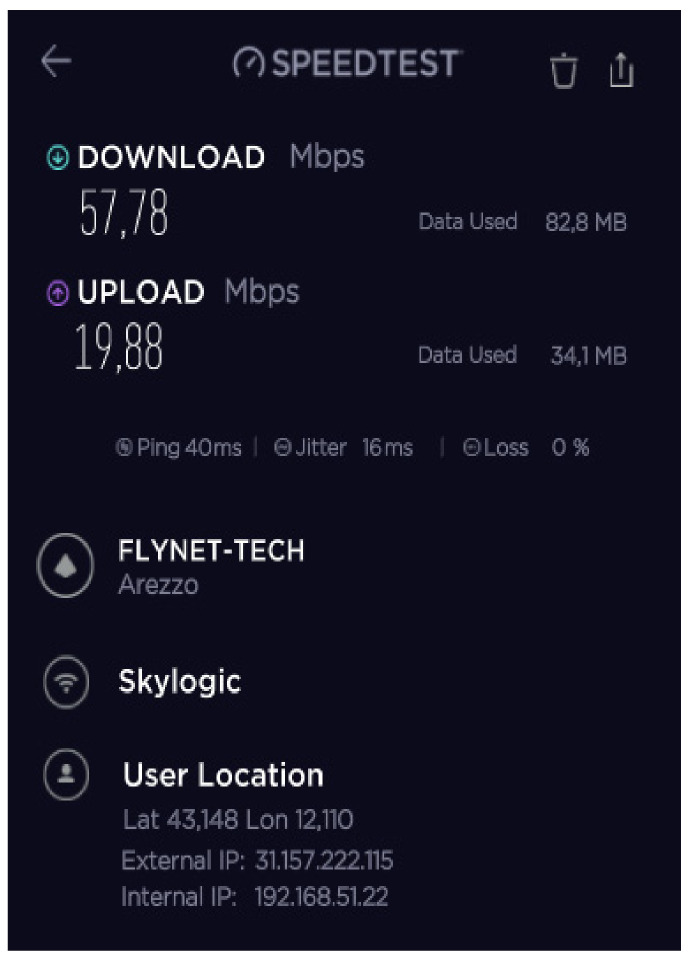
B-LiFE deployment in Turin, Italy, during the COVID-19 crisis: speed test results after combining the SES Scorpion MP80X terminal, 4G/LTE (Vodafone IT) modem, and Eutelsat SatCom (2 kits) using a load balance broadband router.

**Figure 20 ijerph-18-09014-f020:**
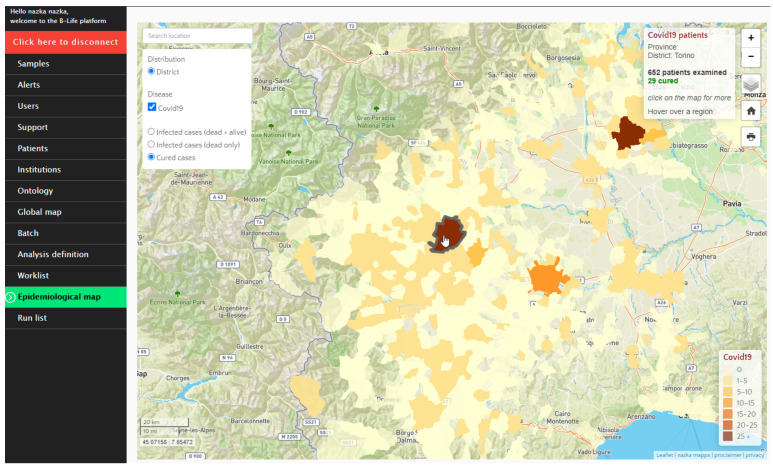
B-LiFE deployment in Turin, Italy, during the COVID-19 crisis: the epidemiological map including the information window on the top right of the map changes dynamically according to the position of the mouse on the map. It shows the name and the number of patients examined in the selected district and the number of infected or cured patients. Although this information was not required during the B-LiFE mission in Piedmont, other types of information, such as survival or death, can be added to the map.

**Table 1 ijerph-18-09014-t001:** Accumulated SatCom data.

Kit	Downlink	Uplink
Rapid Deployment kit:	11.54 GB	4.35 GB
A2C kit	3.05 GB	1.40 GB
Total	14.59 GB	5.75 GB

**Table 2 ijerph-18-09014-t002:** Planning Rx/Tx parameters.

Parameter	TETRA	LTE
Frequency [MHz]	427.8	1800
Nominal bandwidth	25 kHz	5 MHz
Transmitter power	10 dBW (40 dBm) for MS 0 d BW (30 dBm) for HH *-the nominal power for the Exercise–0.01 W	−7 dBW (23 dBm) for both MS and HH
Transmitter antenna gain (including cable) [dBi]	2	8
Transmitter antenna height [m]	3	3
Information type, data rate	Voice	IP traffic 1 Mbit/s
Receiver antenna gain (including cable) [dBi]	0	0
Receiver antenna height [m]	1	1
Receiver sensitivity [dBm]	−106 (dynamic)	−101
Building penetration loss for indoor coverage representative for city buildings [dB]	15	15
Planning margin (PM) for desired coverage [dB]. At least the level [Sensitivity] + [PM] is to be achieved at all points within the desired coverage area	10	10
Planning and presentation level [dBm]	−96 (outdoor) −81 (for indoor margin)	−91 (outdoor) −76 (for indoor margin)

**Table 3 ijerph-18-09014-t003:** Communication range calculation results.

	Indoor	Outdoor
TETRA (0.01W)	0.5 km	1.75 km
LTE	0.5 km	2 km

## Data Availability

No datasets were generated or analysed during the current study.

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
