# Peer review of "Telecommunication Facilities, Key Support for Data Management and Data Sharing by a Biological Mobile Laboratory Deployed to Counter Emerging Biological Threats and Improve Public Health Crisis Preparedness"

_ijerph, 2021, doi:10.3390/ijerph18179014_

Round 1

Reviewer 1 Report

The paper describes different implementations of lab deployments in different countries to support an emergency crisis. The work is well written and depicts the telecommunication system infrastructure required to provide an adequate response to different type of crisis situations.

The transition from section 1 to section 2 can be improved. It was not clear what the paper was about by reading the introduction. I suggest to improve this section to let the reader know about what will be presented in the next section and subsections.

Some figure labels are incorrect (for example: line 185, 241).

Author Response

Answer to Reviewer 1:

Thank you for valuable constructive comments on the paper ijerph-1323301 - Telecommunication Facilities, Key Support for Data Management and Data Sharing by Biological Mobile Laboratory Deployed to Counter Emerging Biological Threats and Improve Public Health Crisis Preparedness.

The paper has been revised and improved accordingly in the following way:

  • The transition from section 1 to section 2 can be improved. It was not clear what the paper was about by reading the introduction. I suggest to improve this section to let the reader know about what will be presented in the next section and subsections.

Section 1 was improved by adding the follow paragraph:

“Based on the PPDR-specific communication needs as presented above, the following section 2 describes the progressive integration of a communication capacity in the B-LiFE deployable (box-based) laboratory. In this section, we illustrate and discuss the integration process implemented through few main field deployments and exercises focusing on rapid response to biological crisis. We explain how efficiency, robustness and sustainability of the integrated telecommunication and box-based laboratory concept was assessed and how it was used to ultimately improve the performance of the B-LiFE team during operational deployments.”

  • Some figure labels are incorrect (for example: line 185, 241).

The incorrect labels have been corrected.

Reviewer 2 Report

This paper discussed the development and benefits of the integration of multi-mission, multi-user nomadic, rapidly deployable telecommunication nodes for emergency uses (TEN)  in the capacity of B-LiFE , which has some theoretical significance. However, the paper needs to be partially improved before publication. My detailed comments are as follows:

  1. In lines 147-148, "New methods and protocols should be described in detail while well-established methods can be briefly described and Words cited." This sentence seems out of context, please confirm.
  2. In lines 522-526,"Figure 18. B-LiFE deployment in Turin, Italy during the COVID-19 crisis: speed test results for the Eutelsat SatCom kits 1 and 2 . The best results were achieved after combining all the aforementioned facilities using a load balance broadband router. The speed test results after this combination are Presented in Figure 19." These three sentences do not match the current context and are the same as sentences 534-538. Please confirm and delete them.
  3. Figure needs to be improved and explained comprehensively.
  4. This article has fewer references, try to enrich.

Author Response

  • Answer to Reviewer 2:

    Thank you for valuable constructive comments on the paper ijerph-1323301 - Telecommunication Facilities, Key Support for Data Management and Data Sharing by Biological Mobile Laboratory Deployed to Counter Emerging Biological Threats and Improve Public Health Crisis Preparedness.

    The paper has been revised and improved accordingly in the following way:

    • In lines 147-148, "New methods and protocols should be described in detail while well-established methods can be briefly described and Words cited." This sentence seems out of context, please confirm.

    These lines are now deleted from the text.

    • In lines 522-526,"Figure 18. B-LiFE deployment in Turin, Italy during the COVID-19 crisis: speed test results for the Eutelsat SatCom kits 1 and 2. The best results were achieved after combining all the aforementioned facilities using a load balance broadband router. The speed test results after this combination are Presented in Figure 19." These three sentences do not match the current context and are the same as sentences 534-538. Please confirm and delete them.

    The incorrect lines were deleted from the text.

    • Figure needs to be improved and explained comprehensively.

    The reviewer does not specify what figure needs to be improved. In any case, all figures and explanations to them have been verified for correctness and clearness. 

    • This article has fewer references, try to enrich.

    “Nine (9) references have been added to the list. As this concept paper is really the first one to focus on the integration of telecommunications tools in a deployable laboratory, we would like to point out that developed technological solution for the optimal combination of a biological mobile laboratory and telecommunication assets, as described in the paper, are unique, and based on our own expertise acquired during the field deployments and research projects from 2009 to 2021. Accordingly, the only references referring to this integration are primarily related to reports of B-LIFE field deployments to financial sponsors (either European Space Agency or European Commission).”

Reviewer 3 Report

The paper enhances the benefit of advanced communication tools in the field of Public Health Crisis Preparedness. The Biological Light Fieldable Laboratory for Emergencies (B-LiFE) tool is such an example, because it has developed a multi-mission, multi-user Field Communication and Control System, which is integrating satellite telecommunication with terrestrial communications (TETRA, LTE and later 5G). Among the many deployable telecommunication nodes for emergency (TEN) are currently used, the authors promote the use of one of such multi-institutions, multi-missions, multi-laboratories (MIML) ICT-toolbox in the area of laboratory information management system (LIMS) called MIML_LIMS. See abstract lines 10-26.

In their 1. Introduction, the authors illustrate the landscape can trigger the involvement of multiple international, European, and domestic stakeholders and coordination mechanisms during a health crisis, such as an epidemic or pandemic outbreak. See lines 32-36.

Fig. 1 at line 37 shows an “Overview of the health crisis stakeholders’ interactions and alert mechanisms at the international, European, and national levels”.

The authors’ interest if directed towards public protection and disaster relief (PPDR) organizations and services are carried out on-site in the field of emergency. See lines 63-69.

Mostly used and developed to this public protection and disaster relief are telecommunications, first of all the radio communication. See lines 70-77.

Fig. 2 at line 96 shows the generally agreed categories of emergency communication, that is: (i) those between authorities/organizations; (ii) those flowing down from authorities/organizations to citizens and back from citizens to authorities/organizations; and (iii) those among citizens. See also lines 78-95.

Section 2. Humanitarian deployments and international exercises is about examples of deployment of assistance and emergency counteracting, such as Paragraph 2.1. Deployment in Guinea related to the case of B-LiFE/B-FAST (Belgian First Aid and Support Team) mission in N’Zerekore, Forest Guinea from December 20, 2014 to March 22, 2015 to fight against the spread of EVD (Ebola viral disease) in West Africa. See lines 99-115.

A description of Emergency.lu C-band satellite terminal provided by SES TechCom and the government of Luxemburg for the B-LiFE mission for three months in N’Zerekore is at lines 128-145. Figure 3. at line 143 provides a view of the satellite communication antenna in N’Zerekore.

Paragraph 2.2. Deployment in Germany provides example in Germany on February 8–12, 2016. Once again B-LiFE was employed this time against waterborne diseases. See lines 149-163.

The authors illustrate particularly the use of the MinION sequencer by Oxford Nanopore Technologies (ONT) in the B-LiFE facility during the exercise Clueless Snowman; a Fig. 4. of the MinION data flow from the laboratory to the tent to the crisis centre is also provided at line 171.

Paragraph 2.3. Deployment in Sweden, describes five exercises for the Medium/Heavy urban search and rescue 204 (USAR), USAR in Chemical, Biological, Radiological, Nuclear (CBRN) conditions held in Revinge, Sweden on the 24–27 April 2017. See lines 202-215.

B-LIFE facilities were deployed under the flag and support of the Belgian First Aid and Support (B-FAST), because following the scenario of this exercise, the southern area of the Faultland region suffered severe flooding. See lines 216-229.

Paragraph 2.4. Deployment in Belgium, shows the Bio-Garden exercise at Level 3 in European security applicable EC countries—justified by an unprecedented wave of terrorist activity in several European cities in the preceding months—was held on June 19, 2018, at the Belgian Defence barracks Major Housiau at Vilvoorde (Peutie), Belgium. See lines 262-269.

Investigations there have allowed to find suspected manipulation and/or production of bioweapons, and suspicious food samples. See lines 290-310.

Fig. 6. at line 286 provides a global view of the of the «Bio-Garden» Exercise.

The generic architecture of a fast deployment telecommunication node for a biological light-fieldable laboratory for emergencies is shown in Figure 7. at line 322.

A more detailed description of the B-LiFE telecommunication team is shown in Fig. 8. at line 335, and at lines 325-370. Fig. 9. at line 371 is an example of Rohill TetraNode equipment (TETRA base station) tested during the exercise, while Fig. 10 at line 404 is an example of Nokia Ultra Compact Network LTE system. Table 2. at line 422 shows parameters both for TETRA and LTE systems. Table 3. at line 425 shows the “Communication range calculation results” for both instruments either indoor and outdoor.

Paragraph 2.5. Deployment in Italy, shows the deployment of B-LIFE from June 11 to July 24, 2020, in Italy, in the Piedmont Region, to bring additional diagnostic test capabilities, including real-time quantitative polymerase chain reaction (RT-qPCR) analysis and antibody (anti-SARS-CoV-2 If M and Ig G) testing. Fig. 11. at line 440 provides an overview of the B-LiFE deployed in Turin accordingly. Fig. 12. at line 460 provides a visual architecture of how communication terminals and equipment have been related each other to interconnect the local system to the internet. Fig. 13. at line 476 is a representation of the core systems used in the critical situation of pandemic. Fig. 14. at line 495 is a representation of the installation of the GovSat Scorpion MP80X SatCom terminal. Fig. 15. is a representation of the installation of EutelScheme 50. Mbps.

Fig. 16. at line 527 is an example of speed test results for the 528 4G/LTE (Vodafone IT) modem, while Fig. 17. at line 530 is an example of speed test results for the SES Scorpion MP80X terminal. Fig. 18. provides at line 533 other two speed test results for the 534 Eutelsat SatCom kits 1 and 2. Fig. 19. at line 539 provides speed test results after combining the SES Scorpion MP80X terminal, 4G/LTE (Vodafone IT) modem and Eutelsat SatCom (2 kits) using a load balance broadband router.

Paragraph 2.5.4. Laboratory Information Management System (LIMS), enters the topic of the tracking and tracing ICT system called a laboratory information management system (LIMS).

The main functionalities needed during this mission were patient data collection, including the physical addresses of the patients for epidemiological mapping, generation of analysis requests (serology [IgG and IgM antibody] and molecular [RT-qPCR] tests), and the generation of sample analysis reports. See lines 559-571.

Paragraph 2.5.5. Epidemiological Maps, shows the four types of maps embedded in the LIMS: (i) global, (ii) epidemiological, (iii) samples/alerts/patients, (iv) mobile-friendly, and the manipulations can be obtained with the aim at summarizing information and borders’ levels. See lines 586-595.

Paragraph 3. Conclusions is about the lessons learned from all the preceding explained missions have used of multi-institutions, multi-missions, multi-laboratories (MIML) ICT-toolbox in the area of laboratory information management system (LIMS).

CHANGE REQUEST:

  • Please eliminate the semicolon after “Mbps” at line 508;
  • I personally would create smaller images to harmonize the distribution of both text and figures.

Kind Regards,

Author Response

Answer to Reviewer 3:

Thank you for valuable constructive comments on the paper ijerph-1323301 - Telecommunication Facilities, Key Support for Data Management and Data Sharing by Biological Mobile Laboratory Deployed to Counter Emerging Biological Threats and Improve Public Health Crisis Preparedness.

The paper has been revised and improved accordingly in the following way:

  • Please eliminate the semicolon after “Mbps” at line 508.

Line 508 is corrected; the semicolon is removed.

  • I personally would create smaller images to harmonize the distribution of both text and figures.

Most of the figures are resized and made smaller, but for some of them it wasn't possible to do so, because they include details and supplementary data which are not readable in small size.

Best regards